# Protocol on establishing a national disease registry–Swiss Pediatric Inflammatory Brain Disease Registry

**Lorena Freya Hulliger** [1] *, **Anne Tscherter**[1], **Claudia Elisabeth Kuehni** [1], **Sandra Bigi**[1,2], **on behalf of the Swiss Pediatric Inflammatory Brain Disease Registry**¶

**1** Institute of Social and Preventive Medicine, University of Bern, Bern, Switzerland, **2** Department of Neuropediatrics, Lucerne Cantonal Hospital, Children's Hospital of Central Switzerland, Luzern, Switzerland

¶ The complete membership of the author group can be found in the Acknowledgments.
* lorena.hulliger@ispm.unibe.ch

**Data Availability Statement:** No datasets were generated or analysed during the current study. All

## Abstract

### Background

Pediatric-onset inflammatory brain diseases are a group of potentially life-threatening central nervous system disorders. Overall, pediatric-onset inflammatory brain diseases are rare and therefore difficult to study. Patient registries are well suited to study the natural history of (rare) diseases and have markedly advanced the knowledge on pediatric-onset inflammatory brain diseases in other countries. Following their example, we established a national pediatric-onset inflammatory brain disease registry in Switzerland (Swiss-Ped-IBrainD).

### Aims

The Registry aims to describe epidemiology, demographics, diagnostics, management, and treatment, since these areas remain understudied in Switzerland. Additionally, we want to promote research by fostering the knowledge exchange between study centers and setting up studies such as national quality of life surveys. We further aim to facilitate the access to national and international studies for patients with a pediatric-onset inflammatory brain disease living and/or treated in Switzerland.

### Methods

The Swiss-Ped-IBrainD is a multicentric, population-based, observational cohort study (IRB number: 2019–00377) collaborating with 11 neuropediatric centers in Switzerland. Patient screening, information and recruitment is mainly conducted by the local principal investigators. The data collection is organized centrally by the Executive Office of the registry. The collected data is purely observational. Medical records are the primary data source. All patients who have been diagnosed with a pediatric-onset inflammatory brain disease since 2005 are eligible. We aim to include all pediatric-onset inflammatory brain disease patients living and/or treated in Switzerland who meet the inclusion criteria. Considering existing literature and our single-center experience we anticipate 300–400 eligible patients.

relevant data from this study will be made available upon study completion.

**Funding:** All funding was awarded to S.B. The project is funded by the following organizations: Swiss Multiple Sclerosis Society (www.multiplesklerose.ch), Roche Pharma (Switzerland) Ltd (award number: 20 f 9-GR-0074; www.roche.ch), Novartis Pharma Schweiz AG (award number: P3 C11912780147; www.novartis.com), and Sanofi-Aventis (Schweiz) SA (www.sanofi.ch). The funders had no role in the study design, data collection and analysis, decision to publish, or preparation of the manuscript.

**Competing interests:** The authors have declared that no competing interests exist.

## Status

Currently, all 11 neuropediatric centers have been initiated and are recruiting. As of the first of May 2023, we have identified 275 eligible participants and obtained informed consent from 101 patients and/or families. None of the informed patients and/or families have refused participation.

## Introduction

Pediatric-onset inflammatory brain diseases (IBrainDs) are rare central nervous system (CNS) diseases with the potential to cause life-threatening conditions. Often, patients experience an acute initial phase with neurological symptoms including focal neurological deficits, cognitive and/or behavioral dysfunctions, seizures, hemiparesis, and ataxia [1, 2].

IBrainDs may be primary (idiopathic) or secondary in origin, e.g., as part of a systemic disease or infection [1–5]. Irrespective of the origin, IBrainD pathology is characterized by immune cell activation, inflammation, cytokine, and/or antibody production. Specific inflammatory pathways target different structures in the CNS [2, 6]. Such structures may be certain segments of blood vessels and neurons or CNS proteins such as myelin, cell surface receptors, channels, or enzymes [6]. Based on either the affected structures or the disease mechanism, IBrainDs may be classified as demyelinating, antibody-associated, T-cell mediated, granulomatous, or CNS vasculitis.

Overall, pediatric-onset IBrainDs are rare. Incidences are only known for a few subgroups, such as demyelinating diseases 0.05–2.85/100'000 persons/year [7–14]. Other IBrainDs are presumed to occur even less frequently. However, epidemiological data, particularly on children, are lacking.

Rare diseases are inherently difficult to study. Their investigation requires special efforts and appropriate instruments. Patient registries are well suited to study the natural history of (rare) diseases; the Canadian pediatric demyelinating disease registry and BrainWorks–a Canadian registry for pediatric IBrainD patients–have markedly advanced the knowledge on IBrainDs in children [2, 6]. Additionally, they have increased awareness and consecutively helped to improve the care of this vulnerable population.

Following their example, a national pediatric-onset IBrainD registry (Swiss-Ped-IBrainD) was founded in 2019.

The primary focus of the Swiss-Ped-IBrainD is on the understudied areas in this field, such as epidemiology, health care management in general (and the diagnostic process in particular), as well as long-term patient outcomes.

Epidemiological data such as prevalence, incidence, or prognostic data are invaluable for recognition of pediatric-onset IBrainDs by the healthcare system. This may favor future policy-making which addresses pediatric-onset IBrainDs.

Pediatric-onset IBrainDs are often diagnosed with a significant delay, which entails a delay in treatment initiation. Optimizing the diagnostic process is one of our priorities since starting treatment early improves disease outcomes [3, 5, 6, 11]–particularly in demyelinating and antibody-mediated diseases. Researching the clinical phenotypes of pediatric-onset IBrainDs and gathering information on the diagnostic workup will help to differentiate the diseases and expedite the diagnosis.

In an acute phase the treatment of IBrainDs is directed at controlling the inflammation. Following the acute intervention, many patients receive specific immunomodulatory

treatment depending on the inflammatory pathways of the underlying IBrainD [6]. Despite appropriate treatment, some children develop functional deficits [3, 11]. We will assess these functional deficits and other outcomes of pediatric-onset IBrainD patients and hope to identify relevant factors that may be implemented in new treatment guidelines.

Furthermore, the registry should enable patients with pediatric-onset IBrainDs to participate in international clinical studies for e.g., new drugs or new therapeutic regimens. Until now, these patients have rarely had the opportunity to participate in international studies–they were at a disadvantage compared to patients from countries with centralized registration such as France or Italy.

Through the Swiss-Ped-IBrainD registry and its linkage with other rare disease registries, we also expect to achieve high enough case numbers for statistically significant results.

## Materials and methods

### Main aims

The Swiss-Ped-IBrainD registry has three main aims.

Firstly, to provide detailed and comprehensive health reporting on pediatric-onset IBrainDs in Switzerland. The reporting includes but is not limited to the following areas of interest: epidemiology, demographics, diagnostics, management, and treatment.

Secondly, to promote research by cultivating a regular exchange between the study centers and providing investigators with access to the high-quality structural data collected by the registry.

Thirdly, to facilitate access to national and international studies for pediatric IBrainD patients living and/or treated in Switzerland.

### Procedure

The Swiss-Ped-IBrainD Registry is a multicentric, population based, observational cohort study. Eleven neuropediatric centers in Switzerland have agreed to participate.

We expect to detect close to 100% of new pediatric-onset IBrainD patients. Since pediatric-onset IBrainDs are severe conditions requiring expert care, most patients in Switzerland will seek help at or be referred to one of the participating specialized centers. Child neurologists in private practices in Switzerland do not treat IBrainD patients themselves [15]; missing new patients with a pediatric-onset IBrainD is therefore highly unlikely.

Patient screening is conducted by the principal investigators (PIs) of the study centers. They compile a screening list and inform all regularly seen, eligible patients of the registry. Patients who do not pay regular visits to the centers (due to various reasons including transition to adult care, moving, and treatment conclusion) may be informed by letter. All informed patients are invited to consent to study participation by means of signing the informed consent form. If the patient is underage, their parent(s)/guardian(s) must also sign the informed consent form. Recruitment has started on the 1st of November 2020. As is typical for a medical registry, recruitment is open-ended and therefore has no fixed end date.

The signed consent form allows us to collect all of the patient's data (including identifying data) as defined in the case report forms. If patients or their families choose not to consent, we can still collect a de-identified minimal dataset including diagnosis, year of diagnosis, year of birth, sex, status (dead or alive) and cause of death, if applicable. This procedure has been approved by the local ethics committee to ensure the completeness of the epidemiological data.

The collected data is purely observational. Medical records are the primary data source.

The collection has retrospective and prospective components. Patient data from the time of diagnosis until present is collected and entered in the registry database (retrospective). Newly

diagnosed pediatric IBrainD patients are recruited continuously. Their data as well as the follow-up data of formerly diagnosed patients are collected and updated regularly (prospective).

The data collection itself is centrally organized by the Executive Office, which is the operational arm of Swiss-Ped-IBrainD and is located at the Institute of Social and Preventive Medicine (ISPM) at the University of Bern. The Executive Office is responsible for the collection of patient data from all participating centers (central data collection). The Executive Office and the PIs work together to maintain a continuous flow of data from the participating centers into the Swiss-Ped-IBrainD database. This includes establishing access to source documents and case report forms for the Executive Office and the transformation of this into a standardized process.

To promote joint research and advance patient management the Task Force on Patient Care and Management was founded. This working group meets bimonthly and is comprised of all PIs as well as additional experts in the field (neuropsychologists and neuroradiologists).

The Swiss-Ped-IBrainD further provides the basis for national surveys, since questionnaire studies in line with the objectives of the Swiss-Ped-IBrainD are covered by the research plan. We are currently preparing the first questionnaire on patient/family perceptions of the diagnostic process and quality of life.

## Study population

Regarding the target population, we aim to include all pediatric-onset IBrainD patients living or treated in Switzerland who have been diagnosed since 2005 (Swiss population $\leq$ 33 years of age: 3'260'287 [Statistik 2010]). Because epidemiological data on pediatric-onset IBrainDs is scarce, the number of eligible patients is merely a rough estimate. Incidences between 0.05–2.85/100'000 persons/year [7–14] have been reported for demyelinating diseases of the CNS and 0.04–0.33/100'000 persons/year [16, 17] for other IBrainDs such as pediatric-onset anti-NMDA receptor encephalitis. Considering these numbers and our single center experience we anticipate 300–400 eligible patients.

**Inclusion criteria.** Patients living and/or treated in Switzerland diagnosed since 2005 with one of the pediatric-onset IBrainDs listed in Table 1.

**Exclusion criteria.** Patients with neurological symptoms due to infectious diseases of the CNS or congenital infections (such as TORCH, peripheral facial palsy), patients with genetic metabolic causes of central demyelinating diseases and patients with neurological symptoms due to Guillain-Barré-Syndrome will be excluded from participation.

## Organization

Swiss-Ped-IBrainD has two main bodies: the Steering Board and the Executive Office.

The Steering Board consists of medical specialists, mainly pediatric neurologists, and has an elected President and Vice-President. It supervises the Executive Office and sets the direction.

The Executive Office, located at the ISPM, consists of the Co-Heads (Swiss-Ped-IBrainD Legal Representative and Clinical Lead), the Project Manager, the Data Manager and other team members. It manages the database, the variables, the archived documents, and the information on organization rules.

Other closely linked structures are the participating clinics and the Task Force on Patient Care and Management.

The main task of the participating clinics is to recruit patients and provide data to the Swiss-Ped-IBrainD. The main task of the Task Force is to discuss complex cases, establish standards of care and work on pressing issues in the field of pediatric-onset IBrainDs. Fig 1 shows the organization of the Swiss-Ped-IBrainD and its associated structures.

**Table 1. Pediatric-onset inflammatory brain diseases to be included in the Swiss Pediatric Inflammatory Brain Disease Registry.**

| Disease Family | Diagnosis |
|---|---|
| Demyelinating | Optic neuritis |
| | Transverse myelitis |
| | Acute disseminated encephalomyelitis |
| | Multiple sclerosis |
| | Neuromyelitis optica spectrum disorders |
| Antibody associated | Anti-NMDA-R |
| | Anti-GAD65 |
| | Anti-AMPA-1/2 |
| | Anti-Lgi-1 |
| | Anti-CASPR-2 |
| | Anti-GABA-1/2 |
| | Onconeuronal antibodies |
| | Hashimoto encephalopathy |
| T-cell mediated | Rasmussen's encephalitis |
| Other | CNS Vasculitis |
| | CNS Sarcoidosis |
| | CNS Lupus |

The list of the included diagnoses might evolve over time given the active research in the field and validation of new techniques for diagnosis.

## Ethics statement

This project was approved by the cantonal ethics committee of Bern (IRB number: 2019–00377) as well as the ethics committees of Northwest and Central Switzerland, Ticino, Zürich, Geneva, Vaud, and East Switzerland, which cover the 11 study centers. It is and will be conducted in accordance with the protocol, the Declaration of Helsinki [18], the principles of Good Clinical Practice [19], the Human Research Act [20] and the Human Research Ordinance [21] as well as other locally relevant regulations. The Swiss-Ped-IBrainD will further adhere to the recent recommendations for health-related registries elaborated by the National Association for Quality Development in Hospitals and Clinics, Swiss Medical Association, Swiss Hospitals Association, Swiss Academy of Medical Sciences, and University Medicine Switzerland [22]. These recommendations serve as basis for quality assurance of registries.

## Data management

We collect and archive all personal, medical, and questionnaire data as well as digitalized documents of the project in a dedicated electronic data capturing system (REDCap ISPM UniBe). The database complies with all requirements for data security and protection.

## Statistical analysis plan

Since the registry has no designated endpoint, we will continuously analyze the collected data and update results according to our evolving objectives.

Initially, descriptive statistics (mean and standard deviation, median and range, summary tables and simple graphics) will serve to characterize pediatric-onset IBrainD patient population of Switzerland. Subsequently, we plan to investigate between-group differences with tests such as t-tests, chi-squared, Wilcoxon-Mann, and Fisher's exact depending on the variables in question.

For more advanced analyses including associations between factors, we will use regression modelling (logistic, linear, Poisson). For follow-up data we will apply statistical methods appropriate for longitudinal data such as cox and Poisson regression or Kaplan-Meier survival curves.

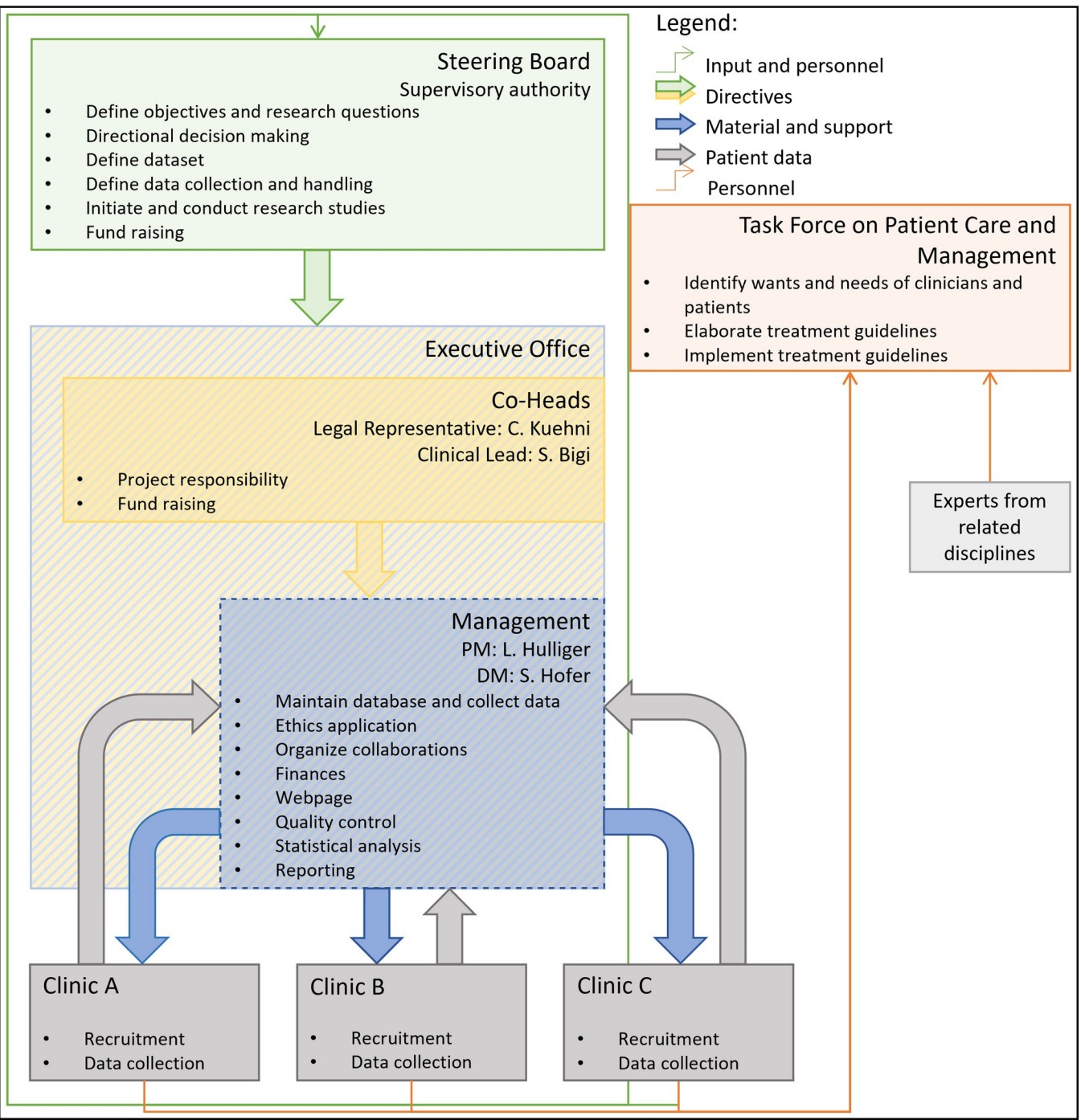

**Fig 1. Organizational structure of the Swiss Pediatric Inflammatory Brain Disease Registry.** PM: Project management; DM: Data management.

We also intend to supplement our datasets with data from relevant routine data and other registries. This allows conducting case-control studies or cohort-based studies about causes and the long-term course of pediatric-onset IBrainDs.

For all analyses we will use STATA (STATA Corp LP, Texas USA) and R (R Foundation for Statistical Computing, Vienna, Austria) or comparable statistical software.

## Safety considerations

The project participation entails minimal risks and burdens (category A). We do not expect any safety issues and therefore refrain from implementing protective measures.

## Current status

Currently, all 11 neuropediatric centers have been initiated and are recruiting. As of the first of May 2023, we have identified 275 eligible participants and obtained 101 informed consents from patients and/or families. None of the informed patients and/or families have refused participation.

## Dissemination plan

Results of this project will be published in scientific journals and presented at scientific conferences. Authorship will be handled according to the ICMJE (www.icmje.org). Ongoing studies and short summaries of the latest news are published on the project website. We will generate annual reports of the Swiss-Ped-IBrainD.

# Discussion

## Potential problems and limitations

We want to reach completeness–meaning we want to collect data from the entire target population. This is challenging since some patients, especially those with mild IBrainDs, may no longer require continued medical care. They are therefore difficult to find and reach (selection/recruitment bias). However, we will exploit all available and legal measures to recruit as many patients as possible. To directly address and minimize the effect of selection/recruitment bias we will collect the minimal dataset from all identified patients after they have been informed.

Misclassification may be a further limitation. To reduce misclassification, we rely on an international classification system and results from biopsies, electrophysiological examinations, and specific laboratory testing (e.g., presence of disease-identifying antibodies such as NMDA-R-AB or NMO-IgG). Patients without a precise diagnosis will not be included. We will also run regular plausibility checks.

Missing data is a further limitation we foresee. To minimize data gaps in the medical history of the patients, we have centralized the data collection; a member of the Executive Office regularly visits the participating centers and collects patient data. These data-collection visits are also good opportunities to resolve open queries and monitor recruitment. Regarding data from questionnaires, we will remind patients to return their questionnaires. On the analysis level, we will use inverse probability weighting to account for an eventual attrition bias [23]. This procedure gives more weight to well-explained observations (by baseline data) and thereby compensates for non-observed follow-up data. Missing data may also be computed by multiple imputation, as done in previous studies [24].

## Amendments and end of project

Substantial changes to the project set-up, the protocol and relevant project documents will be submitted to the responsible institutional review board for approval. This includes the distribution of questionnaires to patients and families.

The project has no defined endpoint. However, the Swiss-Ped-IBrainD may be terminated or suspended due to financial or personnel-related reasons.

## Acknowledgments

We thank Lilianna Bolliger for important ground-laying work. We thank the members of the Swiss Pediatric Inflammatory Brain Disease Registry consortium for their collaboration: Florian Bauder, Andrea Capone Mori, Patricia Dill, Stephanie Garcia-Tarodo, Barbara Goeggel Simonetti, Annette Hackenberg, Judith Kalser, Oliver Maier, Gabriela Oesch Nemeth, Regula Schmid, and Susi Strozzi.

## Author Contributions

**Conceptualization:** Lorena Freya Hulliger, Anne Tscherter, Claudia Elisabeth Kuehni, Sandra Bigi.

**Data curation:** Lorena Freya Hulliger.

**Formal analysis:** Lorena Freya Hulliger, Anne Tscherter, Claudia Elisabeth Kuehni, Sandra Bigi.

**Funding acquisition:** Lorena Freya Hulliger, Claudia Elisabeth Kuehni, Sandra Bigi.

**Investigation:** Lorena Freya Hulliger, Anne Tscherter, Claudia Elisabeth Kuehni, Sandra Bigi.

**Methodology:** Lorena Freya Hulliger, Anne Tscherter, Claudia Elisabeth Kuehni, Sandra Bigi.

**Project administration:** Lorena Freya Hulliger, Anne Tscherter, Sandra Bigi.

**Resources:** Claudia Elisabeth Kuehni, Sandra Bigi.

**Supervision:** Lorena Freya Hulliger, Anne Tscherter, Claudia Elisabeth Kuehni, Sandra Bigi.

**Visualization:** Lorena Freya Hulliger.

**Writing – original draft:** Lorena Freya Hulliger, Sandra Bigi.

**Writing – review & editing:** Lorena Freya Hulliger, Anne Tscherter, Claudia Elisabeth Kuehni, Sandra Bigi.

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
