## [Decision Letter · Decision Letter 0]

26 Oct 2023

PONE-D-23-18512Protocol on Establishing a National Disease Registry – Swiss Pediatric Inflammatory Brain Disease RegistryPLOS ONE

Dear Dr. Hulliger,

Thank you for submitting your manuscript to PLOS ONE. After careful consideration, we feel that it has merit but does not fully meet PLOS ONE’s publication criteria as it currently stands. Therefore, we invite you to submit a revised version of the manuscript that addresses the points raised during the review process.

We look forward to receiving your revised manuscript.

Kind regards,

Sidra Kaleem Jafri

Academic Editor

PLOS ONE

Journal Requirements:

"None of the authors have any potentially competing interest to declare."

6. We note that Figure 1 in your submission contain copyrighted image. All PLOS content is published under the Creative Commons Attribution License (CC BY 4.0), which means that the manuscript, images, and Supporting Information files will be freely available online, and any third party is permitted to access, download, copy, distribute, and use these materials in any way, even commercially, with proper attribution. For more information, see our copyright guidelines: http://journals.plos.org/plosone/s/licenses-and-copyright.

Reviewers' comments:

Reviewer's Responses to Questions

**Comments to the Author**

1. Does the manuscript provide a valid rationale for the proposed study, with clearly identified and justified research questions?

Reviewer #1: Yes

Reviewer #2: Yes

2. Is the protocol technically sound and planned in a manner that will lead to a meaningful outcome and allow testing the stated hypotheses?

Reviewer #1: Yes

Reviewer #2: Yes

3. Is the methodology feasible and described in sufficient detail to allow the work to be replicable?

Reviewer #1: Yes

Reviewer #2: Yes

4. Have the authors described where all data underlying the findings will be made available when the study is complete?

Reviewer #1: Yes

Reviewer #2: No

5. Is the manuscript presented in an intelligible fashion and written in standard English?

Reviewer #1: Yes

Reviewer #2: Yes

6. Review Comments to the Author

You may also provide optional suggestions and comments to authors that they might find helpful in planning their study.

Reviewer #1: I wanted to inquire about some of the process flow regarding recruitment of participants in study which i have mentioned in my comments in word file.

Reviewer #2: The authors may clarify under the Data Availability section whether all data underlying the findings will be made available when the study is complete.

Furthermore, the study period may also be clarified.

7. PLOS authors have the option to publish the peer review history of their article (what does this mean?). If published, this will include your full peer review and any attached files.

Reviewer #1: **Yes: **MUZNA ARIF

Reviewer #2: No

---

## [Author Response · Author response to Decision Letter 0]

26 Jan 2024

PLOS ONE Journal Requirements

Requirement 1.:

Please ensure that your manuscript meets PLOS ONE's style requirements, including those for file naming. The PLOS ONE style templates can be found at https://journals.plos.org/plosone/s/file?id=wjVg/PLOSOne_formatting_sample_main_body.pdf and https://journals.plos.org/plosone/s/file?id=ba62/PLOSOne_formatting_sample_title_authors_affiliations.pdf

Answer:

The manuscript has been changed to meet PLOS ONE’s style requirements.

Requirement 2.:

We note that the grant information you provided in the ‘Funding Information’ and ‘Financial Disclosure’ sections do not match. When you resubmit, please ensure that you provide the correct grant numbers for the awards you received for your study in the ‘Funding Information’ section.

Answer:

The sections ‘Funding Information and ‘Financial Disclosure’ have been changed and are now identical. Please note, that the grant numbers have been added when possible – some grants/donations had no official grant numbers.

Requirement 3.:

Thank you for stating the following in your Competing Interests section: "None of the authors have any potentially competing interest to declare."

Please complete your Competing Interests on the online submission form to state any Competing Interests. If you have no competing interests, please state "The authors have declared that no competing interests exist.", as detailed online in our guide for authors at http://journals.plos.org/plosone/s/submit-now. This information should be included in your cover letter; we will change the online submission form on your behalf.

Answer:

The required statement "The authors have declared that no competing interests exist." has been added to the ‘Competing Interests’ section on the online submission form. The statement has also been included in the cover letter.

Requirement 4.:

PLOS requires an ORCID iD for the corresponding author in Editorial Manager on papers submitted after December 6th, 2016. Please ensure that you have an ORCID iD and that it is validated in Editorial Manager. To do this, go to ‘Update my Information’ (in the upper left-hand corner of the main menu), and click on the Fetch/Validate link next to the ORCID field. This will take you to the ORCID site and allow you to create a new iD or authenticate a pre-existing iD in Editorial Manager. Please see the following video for instructions on linking an ORCID iD to your Editorial Manager account: https://www.youtube.com/watch?v=_xcclfuvtxQ

Answer:

A new ORCID iD has been created and validated in Editorial Manager.

Requirement 5.:

Your ethics statement should only appear in the Methods section of your manuscript. If your ethics statement is written in any section besides the Methods, please delete it from any other section.

Answer:

The ethics statement has been deleted from other sections and appears exclusively in the Methods section of the manuscript.

Requirement 6.:

We note that Figure 1 in your submission contain copyrighted image. All PLOS content is published under the Creative Commons Attribution License (CC BY 4.0), which means that the manuscript, images, and Supporting Information files will be freely available online, and any third party is permitted to access, download, copy, distribute, and use these materials in any way, even commercially, with proper attribution. For more information, see our copyright guidelines: http://journals.plos.org/plosone/s/licenses-and-copyright.

Answer:

Figure 1 has been removed from the submission.

Requirement 7.:

Answer:

The reference list has been reviewed and was found to be complete and correct.

 

Reviewers’ comments

Comment 1.:

Does the manuscript provide a valid rationale for the proposed study, with clearly identified and justified research questions?

Reviewer #1: Yes

Reviewer #2: Yes

Answer:

Thank you for the positive review.

Comment 2.:

Is the protocol technically sound and planned in a manner that will lead to a meaningful outcome and allow testing the stated hypotheses?

Reviewer #1: Yes

Reviewer #2: Yes

Answer:

Thank you for the positive review.

Comment 3.:

Is the methodology feasible and described in sufficient detail to allow the work to be replicable?

Reviewer #1: Yes

Reviewer #2: Yes

Answer:

Thank you for the positive review.

Comment 4.:

Have the authors described where all data underlying the findings will be made available when the study is complete?

Reviewer #1: Yes

Reviewer #2: No

Answer:

Thank you for raising this point.

Since the manuscript does not contain any data besides the number of recruited participants, we feel, that regarding this publication we have met the PLOS Data policy requirement of “all data underlying the findings described in their manuscript fully available without restriction”.

Regarding subsequent publications, we will make as much data available as possible without compromising participant privacy.

Comment 5.:

Is the manuscript presented in an intelligible fashion and written in standard English?

Reviewer #1: Yes

Reviewer #2: Yes

Answer:

Thank you for the positive review.

Comment 6.:

Review Comments to the Author

You may also provide optional suggestions and comments to authors that they might find helpful in planning their study.

Reviewer #1: I wanted to inquire about some of the process flow regarding recruitment of participants in study which i have mentioned in my comments in word file.

Reviewer #2: The authors may clarify under the Data Availability section whether all data underlying the findings will be made available when the study is complete.

Furthermore, the study period may also be clarified.

Answer:

Answer to the comments of Reviewer 1:

• Line number 32,33: 

1.Are you referring to clinical trials here?

The type of trial is unrestricted. But yes, our protocol states, that we can contact the patients/families in the registry for recruitment purposes. This way we can ask our patients if they would like to participate in e.g. international trials that would be of interest to them.

2.For whom the studies will be accessible to and how will you make them accessible?

For all patients/families that have given their informed consent to participate in the registry and who meet the inclusion criteria of the study. E.g., if we are contacted by an institution conducting a study to research paediatric MS (with a formal ethics committee approval), we can collaborate with them and send their informed consent forms and invitation for participation to our patients.

• Line number 89-90:

Is there an estimate how many patients on an average from these French and Italian registries who participated and benefitted from a new clinical trial.It would be nice to know.

We do not have an estimate of how many Italian/French patients have been recruited to participate in clinical trials through the centralized registries.

• Line number 91,92:

Which registries will you link with? European only or other parts of the world. Are there any other Pediatric inflammatory brain disease registries in the world.

We are working on a collaboration with the Dutch registry for demyelinating diseases in children. We intend to link with registries all over the world. But this will take time and a lot of work. There are other paediatric inflammatory brain disease registries such as the Canadian Brain Works. However, the registries sometimes include only one IBrainD, such as paediatric onset MS. 

• Line number 101,102:

Research studies or trials for new drugs?

Both, if possible.

• Line number 107:

How will you ensure that previously diagnosed patients (from 2005 onwards) are not missed as diagnoses of Pediatric inflammatory brain diseases evolve over time.

The possibility that we will miss patients that have been diagnosed as far back as 2005 is high. 

Our priority however, are the patients that have been diagnosed more recently and are therefore still in paediatric care.

We have also already seen a shift of the “diagnoses mix” over time. As an example: the proportion of MOG-AD diagnoses has increased manyfold over the last three years. 

We will disclose this issue and discuss it upon the publication of our first epidemiological data. Thank you very much for raising this point.

• Line number 112:

How will the PI shortlist eligible study participants from screening list.

The PIs of the centres have been trained by the Executive Office to recognise the patients meeting the inclusion criteria. Additionally, the Executive Office checks all patient who consented for suitability.

• Line number 113,114,115:

What will be the procedure of informed consent for such patients whom you are unable of contact/not regularly visiting clinics and you are sending letters to.

We send them an invitation to participate in the registry together with the information letter. If they agree to participation, we may contact them by telephone to explain the proceedings of the registry. If they do not reply after two invitations, we only collect the minimal dataset and mark them as non-responders.

• Line number 117,118:

This recruitment includes new patients or older ones from 2005 onwards.

We also recruit patients who have been diagnosed as far back as 2005.

• Line number 132,133:

How will you sustain the availability of logistics/manpower for continuation of this process at ISPM for long term as the project has no end point.

This is one of our greatest challenges since we are third party funded. Much of our work at ISPM is put into looking for appropriate funders and writing grants. We however have a close collaboration with the Swiss-MS society who is very invested in the project. Furthermore, the Steering Board of the registry supports the acquisition of funds.

• Line number 213,214:

1.are these mostly newly diagnosed patients or older ones?

This depends on the definition of “newly diagnosed”. We found that we have about 20-30 new cases per year. This means, that most of the patients have been diagnosed before the registry has started recruitment.

2.Are Eligible participants shortlisted from screened participants list?

Yes, but the PIs compile the screening lists themselves and are also responsible for the shortlisting.

• Line number 231,232:

This way the older cohort may be missed as definitive diagnostic tests may not have been available previously which are now.

This is indeed a pertinent risk. We will try to minimize the risk as far as possible by checking the diagnostic work-up that has been done at the time of diagnosis (these variables are included in the registry database). In a publication on first epidemiological data, we will discuss this issue and possibly add it as a limitation.

• Line number 233,234,235:

1.How does centralizing the data collection help in recruiting missing data?

The collection of data is essentially done by one person. This allows the implementation of a standardized procedure during data collection. If e.g. a patient changes study centre without notifying the registry, the DM who follows all the patients will notice the change and be able to request the data from both treating centres.

2.Will the executive office member will collect missing data of patients?

If we have a data gap, we first ask the corresponding PI to either send us the missing data if available or provide a statement to their best knowledge. Sometimes however there is a “real” data gap. Where a patient for example left the country for a while, continued the treatment abroad, and then came back. In this case we will try to find a way to extrapolate/impute the missing values and disclose the procedure. 

Answer to the comments of Reviewer 2:

Please see our answer to comment 4.

On the topic of the study period, we are unable to further clarify. The study is set-up as a registry (which is intended to continuously enrol new patients and collect structural data) without a clear end date. Any nested projects, however will have a start and an end date that will be provided in the corresponding publication.

Comment 7:

PLOS authors have the option to publish the peer review history of their article (what does this mean?). If published, this will include your full peer review and any attached files.

Do you want your identity to be public for this peer review? For information about this choice, including consent withdrawal, please see our Privacy Policy.

Reviewer #1: Yes: MUZNA ARIF

Reviewer #2: No

Answer:

Thank you both again for your careful review and valuable input.

---

## [Decision Letter · Decision Letter 1]

2 Apr 2024

Protocol on Establishing a National Disease Registry – Swiss Pediatric Inflammatory Brain Disease Registry

PONE-D-23-18512R1

Dear Dr. Hulliger,

We’re pleased to inform you that your manuscript has been judged scientifically suitable for publication and will be formally accepted for publication once it meets all outstanding technical requirements.

Kind regards,

Sidra Kaleem Jafri

Academic Editor

PLOS ONE

Additional Editor Comments (optional):

Reviewers' comments:

Reviewer's Responses to Questions

**Comments to the Author**

1. Does the manuscript provide a valid rationale for the proposed study, with clearly identified and justified research questions?

Reviewer #1: Yes

2. Is the protocol technically sound and planned in a manner that will lead to a meaningful outcome and allow testing the stated hypotheses?

Reviewer #1: Yes

3. Is the methodology feasible and described in sufficient detail to allow the work to be replicable?

Reviewer #1: Yes

4. Have the authors described where all data underlying the findings will be made available when the study is complete?

Reviewer #1: Yes

5. Is the manuscript presented in an intelligible fashion and written in standard English?

Reviewer #1: Yes

6. Review Comments to the Author

You may also provide optional suggestions and comments to authors that they might find helpful in planning their study.

Reviewer #1: Issues highlighted have been addressed by authors. Authors have answered the queries very comprehensively.

7. PLOS authors have the option to publish the peer review history of their article (what does this mean?). If published, this will include your full peer review and any attached files.

Reviewer #1: **Yes: **Muzna Arif

---

## [Editor Report · Acceptance letter]

5 Apr 2024

PONE-D-23-18512R1 

PLOS ONE

Dear Dr. Hulliger, 

I'm pleased to inform you that your manuscript has been deemed suitable for publication in PLOS ONE. Congratulations! Your manuscript is now being handed over to our production team.

Kind regards, 

on behalf of

Dr. Sidra Kaleem Jafri 

Academic Editor

PLOS ONE